# Therapeutic Plasma Exchange: Current and Emerging Applications to Mitigate Cellular Signaling in Disease

**DOI:** 10.3390/biom15071000

**Published:** 2025-07-12

**Authors:** R. M. Imtiaz Karim Rony, Alireza Shokrani, Naseeb Kaur Malhi, Deborah Hussey, Rachael Mooney, Zhen Bouman Chen, Tristan Scott, Haiyong Han, Jaeger Moore, Jiahui Liu, Wendong Huang, Adolfo Garcia-Ocaña, Maria B. Grant, Karen Aboody, Daniel Von Hoff, Rama Natarajan, Joshua D. Tompkins

**Affiliations:** 1Department of Diabetes Complications, Beckman Research Institute of City of Hope, Duarte, CA 91010, USAdhussey@coh.org (D.H.); rnatarajan@coh.org (R.N.); 2Arthur Riggs Diabetes and Metabolism Research Institute, City of Hope, Duarte, CA 91010, USA; 3Department of Stem Cell Biology and Regenerative Medicine, Beckman Research Institute of City of Hope, Duarte, CA 91010, USA; rmooney@coh.org (R.M.);; 4Center for Gene Therapy, Beckman Research Institute of City of Hope, Duarte, CA 91010, USA; 5Clinical Genomics and Therapeutics Division, Translational Genomics Research Institute, part ofCity of Hope, Phoenix, AZ 85004, USA; 6Department of Ophthalmology and Visual Sciences, University of Alabama at Birmingham, Birmingham, AL 35294, USA

**Keywords:** therapeutic plasma exchange, plasmapheresis, apheresis, blood exchange, inflammation, neurological disease, metabolic disease, diabetes, cancer, extracellular vesicles

## Abstract

Therapeutic plasma exchange (TPE) is a blood purification technique which functions to remove pathological plasma constituents such as autoantibodies, inflammatory cytokines, immune complexes, and extracellular vesicles (EVs) that contribute to a range of disease states. In this review, we examine current and emerging indications for TPE across cardiovascular, metabolic, neurological, inflammatory, and oncological diseases. We cover emerging preclinical animal models and new applications, emphasizing the roles of cellular signaling and EV biology in mediating plasma functions, and discuss unique therapeutic “windows of opportunity” offered by TPE. We conclude that TPE is underutilized in both preventative and precision medicine, and that next generation TPE therapies will involve personalized plasma biomarker and modulation feedback, with synergistic plasma infusion therapies to mitigate age associated disease and promote tissue rejuvenation.

## 1. An Overview of Plasma and TPE Technologies

Plasma, the liquid portion of blood, is ~91% water, and comprises coagulants (fibrinogen), proteins such as albumin and globulin to regulate osmotic pressure, electrolytes and vitamins (sodium, potassium, chloride, etc.), immunoglobulins, and small amounts of circulating hormones and enzymes (Figure 1) [1]. However, several diseases disrupt circulating homeostasis, including both acute and chronic changes to plasma composition, leading to the accumulation and persistence of circulating drivers of disease. The overall clinical intent of TPE is to reduce pathologic plasma components including autoantibodies, immunological complexes, and/or inflammatory cytokines implicated in many disease conditions.

As a refined apheresis technique, TPE involves the extracorporeal separation of plasma components and replacement with donor plasma or saline supplemented with albumin using centrifugation or membrane filtration. Albumin supplementation is associated with lower adverse reaction rates [2]. Briefly, centrifugation TPE involves blood removal, typically sodium citrate anticoagulant addition, and centrifuge-based plasma separation. In membrane-based TPE, blood, usually anticoagulated with heparin, is passed through a hollow fiber filter to selectively remove plasma components, predominately by size. Replacement fluids are infused post-plasma separation or filtration, with calcium being administered before returning to patients to mitigate citrate-induced hypocalcemia (Figure 1) [3].

Historically, TPE can be traced back to 1913 when Abel, Rowntree, and Turner first described their method for extracting and separating blood components in dogs [4]. Though their work was focused on mimicking the natural filtering properties of kidneys, ultimately functioning as a form of dialysis, in course they demonstrated that large amounts of plasma could be removed, as long as red blood cells were replaced. Using “an apparatus of coiled collodion tubes surrounded by a saline solution arterial blood was shunted through these tubes and then returned to the experimental animal’s vein.” The “vividiffusion” apparatus, they concluded, could be useful in managing renal failure. Subsequent developments in anticoagulation (e.g., heparin), and improvements in plastic tubing and sterile connectors, enabled routine plasma donation during the 1950s and early 1960s [5]. Among the first reports of plasma exchange, Schwartz and colleagues used these developments to treat blood hyperviscosity and retinopathy associated with high IgM circulation in Waldenström’s macroglobulinemia patients [6,7].

Modern variations for TPE exist, including double-filtration plasmapheresis (DFPP), which selectively removes high-molecular-weight substances, and immunoadsorption (IA), in which adsorption columns decorated with specific ligands selectively bind and remove plasma factors (Figure 1). For example, lipid apheresis (LA) involves the specific removal of lipoproteins such as low-density lipoproteins (LDL) and lipoprotein(a) [Lp(a)] using a lipid filtrate membrane, effectively depleting this independent risk factor for cardiovascular disease (CVD) [8]. Each technique offers distinct capabilities for disease-specific applications, and combinations exist for selective plasma filtration with integrated immunological sieving to enable precise modulation of pathological plasma components. These plasma separation techniques may be coupled with targeted therapeutic infusions, such as selective immunoglobulin resaturation. An overview of TPE strategies, including complications, which are generally mild and treatable, and potential adjuvant therapies are provided (Figure 1, Table 1). Current and future TPE indications are discussed below.

## 2. Diseases Currently Benefiting from Therapeutic Apheresis

Therapeutic apheresis is well-established for many indications with treatment and TPE frequency guidelines regularly updated by the American Society for Apheresis (ASFA) [9] (Figure 2, Table 1). It remains a cornerstone therapy for thrombotic thrombocytopenic purpura (TTP), where TPE removes large von Willebrand factor multimers that would otherwise bind platelets to form microthrombi, while also allowing the replacement of deficient ADAMTS13 protease through complementary recombinant protein infusions [10,11,12]. In neurological settings, apheresis serves as a frontline therapy for conditions such as GBS, myasthenia gravis (MG), and both acute and chronic inflammatory demyelinating polyradiculoneuropathy (AIDP, CIDP), among others. Here, TPE primary functions to remove autoantibodies and immune complexes that contribute to neuronal damage.

TPE is used as an adjunct to immunosuppressive therapy in acute, life-threatening phases of catastrophic antiphospholipid syndrome (CAPS) and certain vasculitides such as granulomatosis with polyangiitis [9,10,11]. It is also effective for treating systemic lupus erythematosus (SLE) and rapidly progressive glomerulonephritis, often associated with anti-glomerular basement membrane (anti-GBM) disease [9,10,12]. Beyond autoimmune conditions, TPE is utilized in acute liver failure and for the removal of toxins, venoms, poisons, and overt chemotherapy [9,13]. The timing and number of treatments is critical. For example, in Amanita mushroom poisoning, TPE significantly reduces mortality if initiated within the first 24–48 h, with the number of treatments guided by toxin levels and clinical response. In contrast, Guillain–Barré syndrome (GBS) typically requires 3–6 exchanges over 10–14 days, ideally initiated within 7 days of onset (Table 1) [9,14].

TPE is also effective in transplant settings, both for desensitization and to mitigate antibody-mediated rejection. Especially effective for kidney transplantation, TPE is used for both ABO-compatible and incompatible scenarios and is often coupled with IA therapies. TPE is similarly beneficial in liver and hematopoietic stem cell (HSC) transplants, while in lung and heart transplants, it is used for desensitization and antibody-mediated rejection, often in conjunction with extracorporeal photopheresis.

For a list and description of all current therapeutic apheresis indications, ranked by category (Category I–III) of disease responsiveness, readers are referred to Connelly-Smith et al., 2023 [9]. Importantly, this ninth iteration of ASFA guidelines provides several new target indications. Alzheimer’s disease (AD), for example, has been recently designated a category III indication, where the optimal role of apheresis therapy is not yet established, but growing evidence suggests it may hold promise as an individualized therapeutic approach [9,15]. This category III status now extends to idiopathic inflammatory myopathies, including immune-mediated necrotizing myopathies and anti-synthetase syndrome, collectively underscoring the need to decipher not only the underlying mechanisms of these diseases, but also, the systemic and local cellular and biochemical changes in signaling that mediate TPE therapeutic benefits. While case studies offer valuable insights, larger clinical trials, when feasible, will be essential to advance category III candidate diseases to category II status. For clarity, category II conditions are those for which apheresis is considered a second-line therapy, as a standalone, or when coupled with other treatments; whereas category I conditions have the strongest experimental support where apheresis is considered first-line therapy [9].

When considering the list of existing conditions treated with apheresis/TPE, it becomes evident that most interventions were historically designed for reactionary medicine, often implemented to reduce catastrophic levels of circulating antibodies or toxins. However, apheresis benefits may extend to preventing or mitigating a range of chronic disease states, including cardiovascular, renal, metabolic, and neuroinflammatory disorders. Multi-omic-based, predictive, and personalized plasma diagnostics should aid in expanding the spectrum of conditions treated by TPE. This review highlights several emerging indications and the associated cellular signaling pathways involved in disease. We further discuss preclinical TPE models that can accelerate discovery and inform translational human intervention studies. Finally, we emphasize the need to shift toward predictive and preventative TPE strategies to address a broad range of inflammatory and age-associated human disorders.

## 3. New Disease Targets with Potential Therapeutic Responses to TPE

Although TPE is a well-established treatment for an array of recognized indications (Figure 2, Table 1), there are areas where TPE has been used relatively sparingly, in many cases due to the rarity of the condition. However, for some chronic illnesses, TPE use has simply been underexplored, especially in the context of preventative medicine. All studies described are conducted in humans, unless otherwise indicated for preclinical work.

**Table 1 biomolecules-15-01000-t001:** A summary of current diseases/indications treated by TPE with ASFA category (rank), session details, benefits, and complications. * For standard TPE risks refer to Figure 1, common and treatable side effects.

Indication	ASFA Rank	Frequency	Concurrent Therapies	Benefits	Complications	Ref.
TTP (Thrombotic Thrombocytopenic Purpura)	I	Daily, 1/day, until remission (~1–2 wks)	Plasma replacement, steroids, rituximab	Mortality ↓ from ~90% to ~20%; platelet recovery	Catheter events, FFP reactions	[9,16]
GBS (Guillain–Barré syndrome)	I	3–6 sessions total, typically 1/day	IVIG, steroids	Faster motor recovery; improved outcomes at 1 year	Hypotension, hypocalcemia	[9,17]
Myasthenic crisis (MG)	I	4–6 sessions total, 1/day	Immunosuppressants	Rapid muscle strength/respiratory improvement	Hypotension, catheter-related risks	[9,18,19]
Anti-GBM (Goodpasture’s)	I	~14 sessions, 1/day	Cyclophosphamide, steroids	Autoantibody removal; renal/pulmonary stabilization	Standard TPE risks*	[9,20,21]
Catastrophic antiphospholipid syndrome (CAPS)	I	Daily, 1/day for 1–3 weeks	Steroids, anticoagulation	Thrombotic control; antibody removal	Infection, line risks	[9,22]
Hyperviscosity syndromes (e.g., Waldenström’s)	I	5–7 sessions over ~2 weeks, usually 1/day	Rituximab	Viscosity reduction; symptomatic relief	Standard TPE risks *	[9,23]
ANCA-associated vasculitis (rapidly progressive glomerulonephritis)	I	~7 sessions, 1/day	Immunosuppressants (cyclophosphamide, steroids)	Removes pathogenic antibodies; preserves renal function	Standard TPE risks *	[9,24]
Neuromyelitis Optica Spectrum Disorder (NMOSD)	I	5–7 sessions, 1/day	Immunosuppressants	Rapid removal of pathogenic antibodies	Hypotension, line complications	[25]
Multiple Sclerosis (acute relapse)	I	5 sessions over 7–10 days, 1/day	Steroids	Improves symptoms in steroid-refractory relapse	Line infections, hypotension	[9,26]
Cryoglobulinemia	I	~3–8 every 1–3 days, up to ~10 w/DFPP, over 2–4 wks	Antiviral, immunosuppressants	Removal of cryoglobulins	Standard TPE risks *	[27]
ABO/HLA desensitization	II	~3–10+ sessions, generally 1/day	IVIg, anti-CD20	Enables incompatible transplantation; graft survival	Infection, immunosuppression	[9,28]
Hyperlipoproteinemia (familial type)	II	Variable; weekly or biweekly	Lipid-lowering agents	Reduces LDL and triglycerides to prevent cardiovascular events	Vascular access complications	[9,29]
HIT/HITT	III	Variable, often 1/day perioperative	Discontinue heparin, alternate anticoagulants	Rapid anti-PF4 Ab removal; thrombosis prevention	Standard TPE risks *	[30,31]

### 3.1. Neurodegenerative Disease

Circulating systemic factors have increasingly been recognized as modulators of brain homeostasis, particularly in the context of aging and neurodegeneration. While the blood–brain barrier (BBB) restricts the passage of most plasma proteins under healthy conditions, its selective permeability often declines with age and in neurodegenerative states, enabling greater neuroimmune interaction [32,33]. TPE is thus indicated for several neurological disorders, especially those displaying autoantibodies and immune complexes. This shift opens a potential therapeutic window in which systemic plasma modulation may beneficially impact central nervous system (CNS) function.

#### 3.1.1. Current Neurodegenerative Disease Indications for TPE

Established TPE indications in neurology include acute autoimmune disorders such as GBS, myasthenia gravis (MG), and chronic inflammatory demyelinating polyneuropathy (CIDP), where TPE is used to rapidly deplete pathogenic autoantibodies and immune complexes. In contrast, conditions like schizophrenia or multifocal motor neuropathy have shown inconsistent responses to TPE, likely reflecting differing underlying immunopathologies and lack of antibody-mediated mechanisms [34,35,36].

#### 3.1.2. Emerging Neuroimmune TPE Indications

Autoimmune dysautonomia represents a group of acquired heterogeneous conditions characterized by autonomic nervous system dysfunction that are increasingly being considered for TPE. For example, in autoimmune autonomic ganglionopathy, antibodies against ganglionic acetylcholine receptor (AchR) induce autonomic dysfunction and peripheral neuropathy [9]. In postural orthostatic tachycardia syndrome (POTS), patients experience cerebral hypoperfusion and exaggerated “fight or flight” responses, resulting in palpitations and sleep disturbances [9,37]. Ten percent of POTS subjects express AchR antibodies, making them candidates for TPE [38,39,40]. Given the considerable knowledge gaps in the etiology of these diseases, further mechanistic studies are needed to delineate causality versus correlation and identify which circulating factors are most relevant to disease onset and persistence.

AD exemplifies a high-priority, mechanistically plausible but still exploratory target for TPE. In AD, amyloid beta (Aß) plaques and neurofibrillary tau protein tangles are associated with disease progression, for which there remains no effective treatment [9,32,41]. Region-specific BBB dysfunction—especially in the hippocampus and cortex—opens the door to circulating pro-aging factors such as CCL11 and β2-microglobulin. This is especially true given the potential removal of circulating Aß, which is mostly bound to albumin [42,43], and these molecules may modulate neurogenesis and glial activation states. TPE may reduce their systemic burden, but the extent to which this alters CNS parenchymal inflammation remains unclear and likely depends on regional barrier permeability, endothelial transporter activity, and timing. In the AMBAR trial, six sessions of weekly TPE with albumin replacement in patients with mild-to-moderate AD were associated with stabilization of cognitive decline and attenuated brain atrophy (monitoring for one-year) [15]. These results support the inclusion of AD as a Category III indication in the ASFA guidelines, indicating potential but requiring further mechanistic and clinical validation.

Mechanistic insights from preclinical models suggest that systemic plasma exchange may exert central effects not solely through antibody removal, but via modulation of innate immune tone and neuroglial interactions. For example, Mehdipour et al., recently demonstrated that a neutral blood exchange (NBE) in mice, which mimics plasma depletion by using saline washed isogenic donor blood cells, can reduce neuroinflammation by microglia staining, and enhance hippocampal neurogenesis [2,44,45]. The effects appear to be mediated by elevated circulating proteomic content implicated in brain resilience and repair [44] (Table 2). Importantly, these changes occurred without direct pharmacological intervention, suggesting that plasma composition alone can modulate glial state transitions and neurogenic capacity.

#### 3.1.3. Limitations and Opportunities for TPE in CNS Disease

While these findings are promising, it is critical to distinguish systemic from compartmentalized CNS immunity. TPE does not act directly within the CNS but likely influences neuroinflammation through peripheral-to-CNS signaling interfaces. These interfaces vary in anatomical accessibility and cellular responsiveness to systemic inflammatory cues. Table 2 summarizes key neuroimmune compartments through which TPE-modulated plasma components may exert indirect effects on the brain. Most inflammatory processes in neurodegenerative disease occur behind an intact or partially disrupted BBB, and TPE is unlikely to directly access perivascular or parenchymal cytokine pools without adjunctive BBB-modulating strategies. Furthermore, the temporal nature of plasma turnover means that many observed effects may be transient, and chronic neurodegenerative processes may require periodic or combinatorial intervention to sustain therapeutic benefit. To move the field forward, improved preclinical TPE throughput and the use of neurological disease specific models will help identify the most responsive indications for TPE. This will inform and optimize future clinical trial designs. Integration with BBB-permeabilizing techniques (e.g., focused ultrasound) and selective immunoadsorption technologies, as well as the targeted resaturation of key neurological factors (e.g., BDNF), may further refine TPE for chronic neurodegenerative conditions [9,52,53].

### 3.2. Metabolic Disease

Metabolic diseases and obesity are increasingly recognized as long-term contributors to inflammation, cardiovascular, and neurodegenerative diseases, and their global incidence continues to expand. Diabetes, for example, contributes to both macro- and microvascular complications, including retinopathy, and a predisposition to cardiovascular disease and myocardial infarction [54,55,56,56]. Vascular complications and neuropathy are particularly burdensome and can culminate in limb amputation [54]. In metabolic disease, hepatic steatosis is common, and chemical toxicity or substance abuse can drive liver damage [47,57]. Moreover, several inherited metabolic disorders result in liver damage, elevated levels of cholesterol and/or triglycerides (TGs), and ultimately drive chronic multi-system disease.

#### 3.2.1. Current Metabolic Disease Indications for TPE

Metabolic conditions which benefit from TPE include familial hypercholesterolemia (FH) and hypertriglyceridic pancreatitis (HTGP). The use of TPE to treat individuals with FH, particularly in homozygous offspring, highlights the value of TPE in modulating plasma lipid composition. Here, LDL apheresis not only acutely lowers LDL cholesterol (LDL-C) and Lp(a) levels but may also remove Proprotein Convertase Subtilisin/Kexin Type 9 (PCSK9). PCSK9 otherwise binds to LDL receptors and promotes their degradation, meaning fewer receptors are available to clear LDL-C from the blood, and leading to higher cholesterol levels [29,58].

Human trials have demonstrated that regular TPE can stabilize atherosclerotic plaque and reduce major cardiovascular events, with earlier intervention resulting in better outcomes [29]. Beyond lipid and triglyceride-lowering effects, TPE may also transiently reduce circulating levels of advanced glycation end-products (AGEs), oxidized LDL, and inflammatory adipokines, contributing to an atheroprotective metabolic profile [59]. In the context of HTGP, high circulating TGs leads to pancreatic inflammation, often occurring in individuals with existing metabolic dysfunction such as diabetes. TPE has been shown to remove TG-rich lipoproteins and significantly reduce TG content (>50%) after a single session [9]. However, evidence supporting a significant reduction in overall pancreatitis progression by TPE is still limited [60,61].

#### 3.2.2. Emerging TPE Applications in Metabolic Diseases

Though underexplored, emerging evidence suggests a promising future for TPE-mediated metabolic modulation in human therapies. For example, TPE is a frontline therapy for acute liver failure, temporarily substituting liver functions by removing circulating toxins, improving metabolism, and allowing the liver to rest until transplants become available. In autoimmune hepatitis, TPE may limit liver failure by removing pathogenic autoantibodies [9,62,63]. Further, in preclinical NBE/plasma depletion models, there is also a significant reduction in fatty acid deposition and fibrosis in aged murine livers [45].

Despite an early 1983 study indicating plasmapheresis improved C-peptide levels over 18 months—leading to longer partial disease remission, reductions in autoantibody detection, and no increases in insulin requirements—little research has been performed on therapeutic apheresis and Type 1 diabetes (T1D). The authors concluded: “The removal of the patient’s plasma may reduce the number of autoantibodies, complement, or other inflammatory mediators” [64], although the underlying mechanism remains unclear. More recently, a case study indicated that plasmapheresis and mycophenolate mofetil were effective in treating severe insulin antibody-mediated insulin resistance [65,66]. Taken together, early interventions that reduce autoantibody activity and associated inflammation may aid in preserving beta cell function and delaying the onset of T1D. Any significant delay in onset or extension of the “honeymoon phase” of glycemic control would be expected to reduce future incidence of associated complications, such as kidney disease, retinopathy, and cardiovascular disease [55,56,67].

The role of TPE in type II diabetes (T2D) is also largely unexplored, though potential uses for TPE in treating diabetic nephropathy and/or vascular complications have been proposed. Therapeutic effects may be mediated through reductions in AGE products, inflammatory adipokines, and cytokines, though the effect of sustained TPE removal of these factors remains to be determined [48,50,59,68]. With recent developments in preclinical modeling of TPE, Section 4, and larger randomized clinical trials focused on metabolic inclusion criteria, TPE may become important for improving some T2D outcomes as discussed in the subsequent section.

The short- and long-term effects of TPE on circulating metabolomic features have been sparsely profiled but imply kinetic differences. For example, following TPE, glucose and insulin levels typically normalize within minutes to hours, TGs recover by 7 days, and LDL-cholesterol levels typically recover within weeks to months to baseline [69,70,71]. Therefore, TPE may be effective in mitigating several features of metabolic disease yet remains insufficiently explored as a preventative modality in treating chronic metabolic conditions such as diabetes.

### 3.3. Cardiovascular Disease (CVD)

Among complications associated with metabolic disease and diabetes, vascular issues typically arise with poor disease management and prolonged disease duration [55,56,72]. Microvascular and macrovascular complications increase with age and metabolic dysfunction, usually involving endothelial cell dysfunction and leading to hypertension, atherosclerosis, poor wound healing, retinopathy, and increased incidence of ischemic stroke and myocardial infarction [54,55,72,73]. Vascular disease is driven in part by plasma-derived inflammatory lipids, antibodies, and cytokines, which damage endothelial cells and adjacent pericytes, reduce endothelial nitric oxide bioavailability, and narrow or occlude blood vessels [74]. Macrophage and lymphocyte invasion amplifies local inflammatory responses. TPE may confer vascular benefits by normalizing plasma lipid profiles and attenuating inflammatory signaling networks.

#### 3.3.1. Current CVD Indications for TPE

Several CVDs respond beneficially to TPE including thrombocytosis, peripheral vascular disease, idiopathic dilated cardiomyopathy, and vasculitis, among others [9,75,76]. TPE is also effective for immune desensitization during heart transplantation and for treating hyperviscosity syndrome [9]. The latter frequently occurs in cancer patients and is discussed in the subsequent section. In the context of CVD, TPE-mediated reductions to hyperviscosity improves blood perfusion in the capillary beds.

#### 3.3.2. Emerging TPE Applications in CVD

One of the most promising clinical indications of TPE in the context of CVD is its role in managing Heparin-induced thrombocytopenia (HIT), alternatively known as HITT (heparin-induced thrombotic thrombocytopenia)—a life-threatening autoimmune response to heparin therapy. Unfractionated heparin (UFH), which is widely used as an anti-coagulant in most critical surgeries, including cardiopulmonary bypass (CPB), typically elicits non-immunogenic response in patients; however, approximately 8–50% of the UFH-treated patients develop autoantibodies (mostly IgGs) against the complexes of PF4 (platelet factor 4) and heparin (anti-PF4/H Abs) [77,78]. The anti-PF4/H antibodies consequently trigger the pathological activation of monocytes and platelets in 0.5–1% patients via the FcγIIa receptors on the platelets, ultimately leading to platelet aggregation, and procoagulant microparticle secretion [77,79]. Patients’ immune system subsequently clears the aggregated platelets which leads to thrombocytopenia, an early and clinical manifestation of HIT/HITT (>50% drop in platelet counts relative to baseline) [77,80]. In parallel, activated monocytes derive tissue factors and elevated procoagulant microparticles initiate a signaling cascades that establishes a hypercoagulable state which leads to elevated thrombin formation and pathological thrombosis in HIT patients, contributing to ~6–26% of HIT associated mortality [77,79].

Multiple studies argue that pre- or intraoperative TPE sessions could be seamlessly adapted to deplete anti-PF4/H or HIT-antibodies, enabling safer heparin exposure during CVD surgeries. The earliest documented case of preoperative TPE during coronary artery bypass surgery (CABG) in a patient with HIT (anti-PF4/H antibody positive), was reported by Kajitani et al. in 2001 [81]. Later, Jaben et al. reviewed two studies where preoperative TPE on two patients scheduled for left ventricular assist device (LVAD) implantation resulted in no post-surgery complications or clinical manifestation of HIT [82]. Recent reports further corroborate the therapeutic benefits of preoperative TPE to remove HIT-autoantibodies from patients [83,84]. For example, Liu et al. administrated intravenous immunoglobulins (IVIg) prior to LVAD implantation to prevent platelet loss; the patient also received two TPE—one before and one after surgery [85]. Grazioli et al. also reported a case for which patient received three preoperative TPEs prior to LVAD implantation, with IVIG supplementation before and after the surgery, which led to marked reduction of % PEA (PF4-dependent P-selectin expression assay) and SRA (serotonin release assay) [31]. Notably, a single intraoperative TPE can reduce anti-PF4/H antibodies up to 50–80% and halt the development of clinical HIT in patients with a history of preoperative HIT [30]. Altogether, these observations strongly advocate incorporating TPE as an integral part of CVD surgery and postoperative management protocols.

Vaccine-induced immune thrombotic thrombocytopenia, a rare complication occurring post-adenovirus-based vaccination, has been recognized as a new indication for TPE under ASFA guidelines [9]. The condition is characterized by the activation of anti-platelet factor 4 (PF4) autoantibodies, which bind platelet IgG receptors (FcɣIIa), disrupt hemostatic balance, and promote thromboembolic events [86]. With typical disease mortality rates approaching 50%, TPE has been shown to effectively reduce PF4 antibodies and improve survival. In this study, only 2 out of 17 TPE patients died, both of whom had extensive pre-existing intracranial hemorrhage [87].

As described previously, LA reduces apolipoprotein B, lipoprotein(a), and total cholesterol levels, as well as C-reactive protein and fibrinogen, in primary hypercholesterolemia patients, with concomitant major cardiovascular and cerebrovascular event reduction [88]. In patients with chronic limb-threatening ischemia, the most severe form of peripheral arterial disease—particularly those with diabetic foot—15 LA treatments significantly improved wound healing and amputation-free survival [89,90]. Though promising, results from TPE trials highlight a critical need to elucidate the molecular mechanisms underlying therapeutic responses, and to develop more effective preclinical models to expand clinical indications. In rodent models, these efforts have historically included parabiosis, surgically connecting circulatory systems between aged and young or obese and lean subjects. In general, results indicate that aged or diseased mice impart these features to young, healthy parabionts while acquiring multiple beneficial tissue responses themselves [91]. These results are conceptually powerful for human disease, but not directly clinically translatable.

### 3.4. Cancer

A growing body of literature indicates TPE may be effective in certain oncology settings. For example, malignant tumors often secrete pro-inflammatory cytokines, immunosuppressive extracellular vesicles (EVs), and circulating tumor DNA (ctDNA) that can be detected in plasma and influence cancer behavior and systemic responses. These factors can promote immune evasion, metastatic progression, and resistance to chemotherapy or immunotherapy. TPE may offer a means to transiently deplete these tumor-driving factors and render the tumor microenvironment more responsive to therapeutic agents. An overview of TPE roles in cancer therapy is provided (Figure 3).

#### 3.4.1. TPE as a Cancer Supportive Therapy

**Renal protection**. In addition to addressing hypercalcemia and rapidly reducing blood volume, TPE serves an important preventive role in managing multiple myeloma (MM) patients. The removal of immunoglobulin free light chains (FLCs) and the restoration of renal function are critical for treating renal failure caused by cast nephropathy (Figure 3). Although clinical trials have reported conflicting results regarding MM outcomes, MM patients with biopsy-confirmed cast nephropathy appear to benefit significantly from TPE, through FLC depletion [92]. Therefore, TPE extended survival appears dependent on achieving a therapeutic renal response, and recommended only for specific subgroups, such as those with extremely high circulating FLCs or severe renal failure, and not for the routine treatment of MM [9,93].

**Hyperviscosity syndrome.** Among immunoglobulins, IgA, IgG, and IgM are most commonly associated with blood hyperviscosity. The syndrome occurs in 10–30% of Waldenstrom macroglobulinemia patients and 2–6% of MM patients [94,95]. TPE effectively depletes these immunological complexes and improves blood homeostasis [94,95] (Figure 3). This logic has also been applied towards the reduction in triglycerides in individuals with hypertriglyceridemic pancreatitis, but there appears to be limited clinical impact to date [9,96,97]. However, TPE may provide relief to thrombotic microangiopathy and kidney dysfunction associated with pancreatic cancer [9]. Lastly, given the endothelial cell damaging effects from radiation and chemotherapy, TPE should be explored for repairing vascular damage in cancer survivors.

**Cancer-associated autoimmunity.** Myasthenia gravis (MG) frequently occurs with thymomas, a thymic epithelial derived tumor [98]. The condition is caused by autoantibodies against the nicotinic acetylcholine receptor, which functions across synaptic neuromuscular junctions [99]. TPE is highly effective in acute MG exacerbation, and when breathing muscles are affected, myasthenic crisis, working to immediately deplete autoantibodies. Therapeutic effects typically last 3–6 weeks and routine TPE represents an option for long-term MG management [9,18,19,38].

**Removing excess chemotherapy and cytokines.** TPE is effective in removing overt chemotherapy exposure (Figure 3). For example, TPE has been used to treat acute cisplatin toxicity [76,77]. TPE has also been used to remove long half-life of vascular endothelial growth factor (VEGF) inhibiting Bevacizumab, enabling surgery without wound complications [78]. TPE is also effective in treating cytokine release syndrome associated with chimeric antigen receptor T-cell (CAR-T) therapies [79], and for reducing immune-related adverse events to immune checkpoint inhibitors [80]. Though acute cisplatin toxicity and some monoclonal antibody therapies are effectively removed by TPE, various drugs display unique retention and clearance properties [13,100]. This has important implications for virtually all medications patients may take prior or during TPE courses and have been summarized in an excellent review by Mahmoud et al. [101].

#### 3.4.2. TPE as a Cancer Intervention

There is some evidence that therapeutic apheresis using a B2-microglobluin filter can reduce circulating tumor cell counts, which is expected to reduce metastatic potential [84,85]. In metastatic melanoma specifically, soluble PD-L1 (sPD-L1) and extracellular vesicle (EV) forms of PD-L1 mediate immune checkpoint inhibitor resistance (Figure 3). This occurs through cancer cell presentation of PD-L1 and its suppression of T cell function through PD-1 interactions [102,103]. TPE removes PD-L1 barriers and resensitizes cells to checkpoint inhibitors [103,104,105].

Given the aforementioned vascular effect of TPE, one may be concerned about feeding tumor growth through new blood vessel formation, although clinical evidence for enhanced tumor vascularization from TPE is currently lacking. While the elimination of tumor-driving factors is of clear strategic benefit, this should be performed without removing anti-tumor immune surveillance. Thus, refined tumor-targeted plasma filtering and/or targeted immune system resaturation represent promising future adaptations of TPE, with potential to reduce treatment-related risks and enhance TPE-mediated cancer interventions.

Lastly, a variant of TPE, extracorporeal photopheresis (EC), involves ex vivo leukocyte exposure to 8-methoxypsoralen and UVA light, before reintroducing treated cells. This is highly effective for treating cutaneous T-cell lymphoma [106,107]. Here, corresponding EC malignant cell apoptosis results in tumor cell antigen presentation following phagocytosis and stimulates Th1 dendritic cell targeting of tumor cells.

#### 3.4.3. Cancer Metabolism and TPE

It is important to note that most cancers switch to glycolytic metabolism, rather than oxidative phosphorylation, to support energy demands of rapid growth, even in the face of mitochondrial dysfunction (Warburg effect) [108] (Figure 3). Consequently, a number of effective cancer drugs inhibit glycolysis or glucose uptake. Glycolysis itself generates H+ ions and lactate, and both acidosis and lactate support cancer growth through increased angiogenesis and reduce immune cell infiltration [108,109]. These effects occur particularly within the tumor microenvironment and involve lactate driven M2 macrophage polarization leading to the suppression of anti-tumor immune responses and increased metastasis. Interestingly, TPE was shown to significantly reduce lactic acid levels in septic shock patients at 24 h and significantly improved patient survival [110]. Thus, it is possible that routine TPE treatments could mitigate tumor acidosis.

As described in the Metabolic Disease section, similarly to lactate, most peripheral blood metabolites are expected to be variably depleted by TPE. Therefore, cancer-specific TPE vulnerabilities likely exist for other metabolic pathways, including glutamine, fatty acid, and nucleotide metabolism pathways [111]. Yet, comprehensive profiling of the metabolic effects of TPE in cancer contexts, including differences in metabolite clearance and reconstitution, is currently lacking. As such, metabolic mitigation of cancer development and metastasis through TPE represents an important treatment paradigm for continued exploration, especially for TPE-mediated effects within tumor microenvironment and future refinements to TPE-mediated modulation of cancer-driving metabolites may offer therapeutic windows of opportunity to target cancer vulnerabilities during metabolic shifts.

Overall, there is growing evidence that large-scale plasma manipulation may mitigate tumor growth and metastasis, at least in some contexts. The strategic depletion of tumor-driving factors and infusion of tumor-targeting agents will be central to these efforts. It is important to note that for many TPE patients, frequency and treatment duration are often individualized, and future TPE therapies will likely need to be tailored to cancer type and stage, chemotherapy dosing and clearance kinetics, as well as the patient’s age, sex, immune profile, and metabolism. Real-time monitoring of tumor responses using plasma and TPE-associated diagnostic signatures, and larger-scale early intervention trials, should provide future breakthroughs in TPE-mediated cancer therapies.

## 4. Preclinical Modeling of TPE

For preclinical apheresis modeling, relative to mice, rats have been favored due to their larger blood vessel sizes. For example, a membrane filtration TPE approach in rats using carotid and internal jugular vessels, moving fluids with a peristaltic pump has been described by Ciorpac et al., 2023 [112]. Further, current developments have opened up TPE-like blood exchanges in mice as well [45,113]. Pioneered by Conboy et al., a single heterochronic blood exchange (HBE), which effectively dilutes circulating plasma similar to TPE, was shown to improve skeletal muscle injury recovery in aged mice, reduce fatty liver and fibrosis, and improve neural stem cell proliferation [114]. The term heterochronic refers to the use of young donor cells in aged mice, and subsequent studies have since shown similar benefits during isochronic blood exchanges (aged-aged exchanges) [45]. These studies collectively indicate that removing the inhibitory effects of aged blood plasma appears through depletion outweighs the benefits of youthful blood cell infusion, however the effects on vascular responses in this model are unknown.

Though more clinically translatable, the murine blood exchange procedure is challenging and time-consuming, resulting in low experimental throughput. To overcome these limitations, we have recently developed an automated syringe pump-based exchange platform to process multiple mice simultaneously (Figure 4A). After triple blood exchanges, aged diabetic mice with ischemic injury displayed significantly improved blood flow recovery compared to sham-exchanged mice (Figure 4B). These results emphasize the therapeutic potential offered by TPE in treating vascular damage and promoting blood flow perfusion in metabolic and age-related disorders. Thus, future clinical trials should explore the application of TPE to mitigate endothelial cell dysfunction associated with chronic metabolic disease and inflammation. As compared to human TPE, murine blood exchanges fall somewhere between membrane- and centrifugal-based TPE in terms of total plasma removed (Figure 1). By equalizing the volumes between the recipient mouse and saline washed, plasma depleted, donor blood cells within syringe, ~50% plasma depletion is achieved during a single session. Though we find repeat sessions are tolerated well in mice, unlike humans, mice must be anesthetized during the procedure, and therefore sessions must be separated by a resting period of usually at least one day. Typical human TPE sessions occur over 1–4 h, comparable to the ~90–120 min required for murine exchanges, depending on flow rate selected. This usually involves a flow rate of up to 90 μL/min for the jugular vein using 250 μL infusion/withdrawal volumes, and short 30 s resting intervals between cycles for syringe mixing. Other differences include the inability to utilize peripheral veins in mice due to their small size, and current limitations for effectively titrating anticoagulants.

**Figure 4 biomolecules-15-01000-f004:**
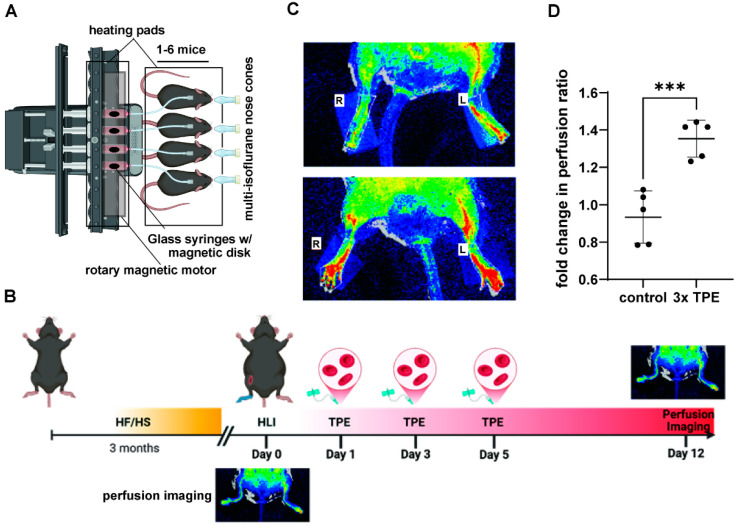
Automated murine TPEs and associated improvements in hind limb ischemia (HLI) recovery. Automated triple plasma exchanges (TPE) in aged diabetic mice improve ischemic vascular injury recovery and blood. (**A**) Overview of automated murine TPE. Briefly, mice with jugular catheters and vascular access buttons are anesthetized and their blood exchanged with saline washed donor blood cells within glass syringes containing magnetic stirrers. Using an equivalent donor fluid volume to that of the recipient mouse and through a series of automated infusion and withdrawal cycles, blood exchanges effectively dilute recipient mouse plasma by ~50% over 2 h. (**B**) Experimental timeline. 80-week-old male mice fed high fat high sucrose (HF/HS) for 3 months underwent HLI on the right limb (femoral artery ligation). TPE or control mice (all surgeries, sham exchange) were performed on day 1, 3, and 5 post ligation. (**C**) Representative images of perfusion on day 12 post-ligation in control and triple-exchange group. Perfusion was measured by LAser Speckle Contrast Analysis (LASCA), left and right legs noted by L and R, respectively. Red coloration indicates increased perfusion. (**D**) Quantification of fold change in perfusion pre- vs. post-TPE/sham exchange. (N = 5, *** *p* = 0.0006, Student’s *t*-test).

Recently, an acoustofluidic-based apheresis system has also been described. Using a fluid stabilizer array on a chip allowed users to separate blood components using low blood volumes (e.g., extracorporeal volume of 280 μL in mice) [113]. This microapheresis system was effective in separating blood into plasma and cellular components, as well as antibodies in a continuous manner. However, the separation of antibodies and cells was incomplete, and the effects on many soluble plasma components, such as metabolites remain to be determined. Taken together, new advances in automated small volume blood exchanges and apheresis technologies are finally opening the door to advancements in preclinical modeling of human TPE, which should facilitate the expansion of conditions treatable by TPE, and perhaps aid in developing more effective TPE treatments in small children and infants [115].

## 5. Cellular and Molecular Signaling Effects Associated with TPE

Several interconnected inflammatory cells signaling networks drive aging and chronic illness. Here, natural responses to cellular stress, infection, and/or tissue damage become dysregulated over time, and promote an underlying low grade systemic inflammation, so called “inflammaging.” Examples of key signaling components include persistent activation of nuclear factor kappa-light-chain-enhancer of activated B cells (NF-κβ), improper janus kinase/signal transducer and activator of transcription (JAK/STAT), NOD-, LRR-, pyrin domain-containing protein 3 (NLRP3) inflammasome, activation of Toll-like receptor (TLR) signaling, and dysregulated mitogen-activated protein kinase (MAPK) networks, among others [35,116,117,118,119]. Local and systemic interactions between these pathways reinforce long-term inflammatory states, drive tissue damage, improper immune responses, and underly chronic illness and age-related decline (Figure 5). For example, cytokine IL-1 is released by a range of cells, including macrophages, monocytes, fibroblasts, and dendritic cells in response to infection, chronic illness or stress, or injury [96]. NF-κβ activation by cytokine IL-1 drives upregulation of inflammatory transcripts which promote immune cell recruitment, while NF-κβ simultaneously inhibits JNK/ERK components of MAPK signaling to prevent caspase 3 driven apoptosis of damaged cells, elevating inflammation and tissue damage further [120]. STAT3 is also activated in response to NF-κβ-mediated cytokine release, and in lymphoma promotes p52 and CD30 expression to sustain non-canonical NF-κβ activation [121]. Several STAT3 target genes promote a feedforward loop between normal and transformed cells, promoting cancer driving inflammation, angiogenesis, and metastasis, while disrupting cancer surveillance (Figure 5) [102,122]. Therefore, therapeutic modulation of cell signaling networks involved in inflammaging represents a promising strategy to combat such decline and promote a more enduring form of healthy aging.

### TPE-Mediated Changes to Inflammatory Signaling

It is well known that inflammatory signaling increases with age and most chronic illnesses. By removing circulating cytokines, complement proteins, and immune complexes, TPE can attenuate both innate and adaptive immune system activation, and thus represents a powerful tool in mitigating systemic inflammation [2].

In early reports from patients with polymyositis, TPE was demonstrated to enhance monocyte function [123], decrease natural killer, B, and T cells, and increase CD4 + and CD8+ T cells [124]. While in acute inflammatory demyelinating polyneuropathy, B cells were diminished, with increases in T cells and regulatory T cells (Tregs) [125]. In systemic lupus erythematosus (SLE), Treg levels are low relative to healthy controls, but rebound to normal levels post TPE [126]. TPE is also associated with a shift in T helper cells (Th) from Th1 to Th2. Th2 cells promote humoral immunity through B cell development rather than cell-mediated immunity. Evidence for long-term cytokine removal, however, is mixed [50].

In pre-clinical TPE studies in aged mice, exchanges were associated with transient depletion of most immune cell types, reduced *immunoglobulin production*, with increases in naïve T cells and increased *erythropoietin signaling* one week post exchange (manuscript in preparation, RNA-seq, significant GO terms in italics). Overall, TPE transiently re-wires the fundamental composition of the immune system. Decreased NK activity, Th2 shifts, improved monocyte function, and elevated Treg counts, coupled with transient reprieve from circulating immune complexes, should reduce many inflammatory drivers of aging and diseases. This represents a more favorable circulating environment for elevated erythropoiesis and vascular repair (Figure 4B). Additional discussion of inflammatory signaling can be found in Effects of TPE on Extracellular Vesicle-Based Drivers of Disease (below).

## 6. Evidence for TPE in Tissue Rejuvenation

Evidence for TPE-mediated tissue rejuvenation or regeneration in humans is lacking. However, promising insights can be gleaned from preclinical murine work. A single “neutral blood exchange,” which in course dilutes plasma ~50%, was associated with improved skeletal muscle regeneration following cardiotoxin injury in aged mice [32]. Livers also appeared less fibrotic and exhibited reduced adiposity. Exchanges were also associated with increased neural stem cell proliferation in the hippocampus, which may relate to cognitive benefits observed in a recent human TPE AD trial [15,20], and perhaps could be extended to other forms of cognitive deficits such as those associated with autoimmune encephalitis [127]. In the same report, aged human serum was inhibitory to primary myoblast proliferation, yet post single TPE, serum associated cell cycle inhibition was removed, and robust myoblast division became evident [45]. Proteomic analysis from a related TPE clinical trial indicated a “molecular re-setting”, with TPE-mediated upregulation of proteins involved in tissue repair and immune function [45]. Higher throughput testing and disease specific modeling of TPE effects will aid in understanding the limits of TPE-mediated tissue rejuvenation.

For human therapies, larger clinical trials with defined endpoints in tissue repair are essential but currently lacking. Yet, routine blood depletion appears to be associated with changes in HSC behavior, improved vascular profiles, and reductions in inflammatory signaling (Figure 4) [88,128]. Such responses may be useful in many contexts of vascular dysfunction, including CVD and metabolic disease, but also for surgery, chemotherapy, or radiation associated tissue damage in cancer patients. Here, the temporal modulation of circulating atherogenic factors, including the modulation of the immune environment away from inflammation, and towards improved vascular function, may promote multi-organ rejuvenation.

Other mechanisms by which TPE may promote tissue regeneration include the depletion of immunosenescence and senescence-associated secretory phenotype (SASP) signaling [2,49], the temporal modulation of factors which control organ size such as Hippo signaling [129,130,131], and the re-activation of adult stem cell niches as discussed next.

### Effects of TPE on Stem Cell Niches

Aging and chronic inflammation erode the regenerative potential of stem cell compartments by altering their microenvironment. Indeed, most stem cells, including HSCs, NSCs, and mesenchymal stem cells (MSCs) all exhibit reduced self-renewal and lineage fidelity over aging [132]. This is likely in part due to stem cell responses to senescence-associated secretory phenotypes (SASP) components, found in circulation, originating from senescent cells, and promoting the propagation of this behavior [46,49,133]. In heterochronic parabiosis and TPE mouse models, re-engagement of stem cell activity is evident in the aged mouse. Here, HSCs are particularly sensitive, with upregulation of genes involved in neutrophil activation and antimicrobial responses, and downregulation of those linked to cytokine production, blood cell formation, chromatin structure, circadian rhythm, and cell cycle control [134,135]. Overall, transcriptional profiles of aged long-term HSCs (LT-HSCs), short term HSCs (ST-HSCs), and multipotent progenitor cells (MPPs) more closely resembled those of young control mice following heterochronic parabiosis, with reduced myeloid bias. Key transcription factors involved in this process include elevated levels of activating transcription factor 3 (Atf3), which protects against stress-induced depletion of hematopoietic stem cells (HSCs), and activating transcription factor 4 (Atf4), which promotes HSC repopulation [134].

Additional mechanistic insight may be gleaned from studies from high frequency blood donors. Karpova et al. recently examined 217 older male blood donors who had donated >100 times their lives. There were no increases in clonal hematopoiesis in these patients, rather, high donor frequency associated variants in DNA methyltransferase 3A (DNMT3A) improved erythropoietin (EPO) responsiveness, and erythroid lineage bias in HSCs [128]. Vascular improvements mediated by TPE may also help restore proper hypoxia homeostasis in stem cell niches (Figure 4).

Parabiosis models and studies of stem cell maintenance and differentiation clearly imply Wingless—related integration site (Wnt) and Notch signaling as key mediators of stem cell activation and tissue regeneration. For example, activation of Notch, and inhibition of Transforming Growth Factor β and phospho-Smad3 signaling (TGFβ/pSmad3) and Wnt signaling improved regeneration of aged muscle stem cells [136]. Inhibition of TGF-β signaling is important for NSC expansion in aged mice, as well as chemokine signaling [137]. Chemokine CCL11 was associated with impaired NSC activity and neurogenesis, a property transferable to young mice treated with CCL11 [44]. Thus, plasma depletion of such molecules may be associated with increased murine hippocampal NSC proliferation, and improved learning and memory [44]. The extent of which routine TPE can enhance stem cell growth and maintenance pathways, and mimic long-term parabiosis effects, remains to be determined.

## 7. Effect of TPE on Extracellular Vesicle-Based Drivers of Disease

Among circulating plasma components, circulating extracellular vesicles (EVs) have been recently identified as major signaling components of inflammation and disease. EVs represent a distinct class of nano-sized, membrane-enclosed, cell-secreted bioactive particles containing cargo components (e.g., miRNA, proteins, metabolites, signaling molecules or peptides). They are secreted by virtually all cell types and usually contain cargoes reflective of their cell of origin, and the relative state of cellular health. Minimal information for studies of extracellular vesicles (MISEV2023) guideline categorizes EVs smaller than 200nm as small extracellular vesicles (sEVs) and those greater than 200nm as large EVs (lEVs) [138]; as a result, classical exosomes are now referred to as sEVs [139]. sEVs are generated via the endosomal pathway where multivesicular bodies (MVBs) fuse with the plasma membrane and releases sEVs in the extracellular space where these vesicles represent primary mediators of intercellular communication, functioning as drivers of organismal development, and long-range communication across organ systems [138,139,140]. Unless otherwise specified, when referring to EVs we are referring to sEVs, also known as exosomes.

To date, EVs have been collected from many tissues, cell sources, and disease states, and, in general, cells involved in tissue repair, such as MSC EVs, secrete EVs with strong therapeutic potential. On the other hand, EVs circulating in disease, or from dysfunctional tissues, often carry disease perpetuating cargos (Figure 5) [103,138,139,140,141,142]. This occurs as affected or damaged cells/tissues tend to release EVs in their extracellular milieu, and into circulation, stimulating immune responses, and amplifying the damage/disease signals. Plasma EVs can thus serve as biomarkers of disease and have been implicated as drivers in diseases ranging from pathological pregnancy [141] to myocardial infarction [143], neurodegenerative disease [51], and cancer metastasis [142].

### 7.1. EVs in Alzheimer’s Disease (AD)

Although AD pathophysiology and underlying mechanisms have been rigorously investigated, recent studies have shed new light on the implications of EVs in AD. Aharon et al. demonstrated that plasma EVs isolated from AD patients show greater proportion of endothelial cells-derived EVs and elevated levels of myelin oligodendrocyte glycoprotein (MOG), as well as pro-inflammatory cytokines IFNγ, chemokine (C-C motif) ligand 5 (CCL5), and GRO (growth-regulated oncogene-alpha, CXCL1) compared to health controls [144]. Analyzing a cohort of early-onset mild cognitive impairment patients, Cano et al., identified proteomic signatures (184 proteins) of plasma EVs that strongly correlate with cerebrospinal fluid (CSF) p-tau181, brain MRI-based imaging of Aβ accumulation, and Mini-Mental State Examination scores [145]. Thus, circulating EVs offers tremendous potential in establishing markers of early onset neurodegenerative disease.

Much of neuronal tau is packaged into the EVs and EVs-associated tau is phosphorylated at Thr-181 (AT270)—a canonical biomarker of AD [144,145,146,147]. Interestingly, EVs isolated from mild/moderate human AD patients’ CSF contain significantly higher proportion of phosphotau compared to aged-matched controls [147]. These brain derived EVs are also responsible for spreading AD-associated tau pathology to other neurons [146], and pharmacological inhibition of EVs biogenesis in a murine model of tau pathology exhibited reduce tau propagation, both in vitro and in vivo [148]. Systematic depletion of pathological EVs could attenuate, and perhaps reverse AD progression, but for now, TPE for AD is considered a recent category III indication by ASFA guidelines [9]. Given the AMBAR trial has shown cognitive benefits for TPE with albumin treated AD patients [15], we expect wider utilization of this promising therapeutic approach, and opportunities to assess TPE clearance of EVs implicated in AD pathogenesis.

### 7.2. EVs in Metabolic Disease

A growing number of studies have revealed that EVs play a major role in the pathogenicity of T2D and T1D [149,150]. For instance, T2D patients exhibit elevated levels of circulating plasma EVs compared to euglycemic patients and in vitro experimental data suggest that hyperglycemia-induced insulin resistance is likely responsible for elevated plasma EV secretion [151]. Interestingly, EV cargo composition, particularly those pertaining to insulin signaling, e.g., leptin receptor and phospho-IR (phosphorylated insulin receptor), is altered in T2D EVs [151]. Plasma EVs from T2D patients also exhibits increased levels of pro-inflammatory proteins CCL28, CD40, and TWEAK, relative to non-diabetics [152], and plasma EVs from diabetic mice induce hyperpermeability of endothelial cells in vitro, and enhanced cerebrovascular leakage in non-diabetic mice in vivo [153].

With regard to T1D, Cianciaruso et al. reported that both rat and human islets secrets EVs that contain GAD65, IA-2 and insulin/proinsulin autoantigens, hallmark biomarkers of T1D development in human [154]. Islets’ EVs are internalized by antigen-presenting cells such as dendritic or B cells, leading to their stimulation in vitro [154]. In fact, conditioning islets with pro-inflammatory cytokine (IL-1β and IFN-γ), mimics the inflammatory milieu of T1D, and induces 2–3-fold higher EV production. Thus, EVs likely serve as major carriers of T1D autoantigens and under inflammatory stimuli, EVs from islets may aggravate islet inflammation and β cell dysfunction. To this end, Dekkers et al., have reported that human β cells under ER stress, a common feature of T1D, generates EVs which promote primary human monocyte activation [155]. CD4+ T lymphocyte EVs enriched for miR-142-3p, miR-142-5p, and miR-155, upregulate cytokine and chemokine signaling, including Ccl2, Ccl7, IFNγ, and Cxcl10 (T1D associated chemokines), and promote apoptosis in treated β cells [156]. Systematic depletion of such pro-inflammatory plasma EVs could alleviate the progression of inflammation associated metabolic disease and perhaps promote the recovery of diabetes associated vascular dysfunction; however, the durability of these effects remains to be determined.

Lastly, EVs have been shown to carry cholesterol and interact with circulating lipids, such as LDL. Indeed, LDL particles are often co-isolated with EVs and can interfere with standard EV diagnostics such as nanoparticle tracking analysis, potentially overestimating EV numbers [157,158]. LDL can augment EV associations with some target cells, both positively and negatively [157,158]. Thus, it remains unclear how hypercholesterolemia and hyperlipidemia syndromes contribute to EV functionality and their propensity to be removed by different apheresis techniques. Alterations to natural biodistribution and EV uptake are expected in many aging and disease contexts, and EV tracking technologies are improving to delineate contact-mediated EV signaling from cargo delivery. Plasma co-purification strategies which remove target lipids (LA) and select metabolic disease driving EVs represent future therapeutic goals, and areas for much needed investigation.

### 7.3. EVs in Tumor Progression

It is now recognized that tumor cells actively secrete EVs to communicate with neighboring cancerous and non-cancerous cells; tumor EVs cargo is enriched with ranges of oncoprotein and tumor-promoting growth factors, which enhance tumor growth, promote neovascularization, immune evasion, and facilitate local invasion [159]. For example, elevated tumor PD-L1 EV levels enable tumor cells to suppress T cell activation through PD-1 to PD-L1 interactions, contributing to T cell exhaustion [160,161,162]. Clinical outcomes of cancer immunotherapy with immune-checkpoint inhibitors (ICIs) for non-small cell lung cancer appear to be highly correlated with level of plasma PD-L1-EVs, in which non-responders show elevated level of plasma PD-L1-EVs compared to responders [163]. Recently Orme and colleagues demonstrated TPE was successful in removing a soluble splice variant of PD-L1(sPD-L1) secreted by malignant cells as well as PD-L1 EVs from melanoma patients’ plasma, with each TPE session removing approximately 71% sPD-L1 and 33.5% PD-L1-EVs, respectively [104]. These effects were extended to a recent phase I clinical trial in which TPE and radiation treatments were found to resensitize immunotherapy-refractory melanoma [105]. Here, overall sPD-L1 was reduced by 78% with circulating sPD-L1 predicting overall survival. These observations strongly argue for TPE in contemporary cancer therapy regimes, aimed at reducing tumor driving EVs, especially those containing well established and TPE depletable disease biomarkers such as PD-L1 (Figure 5). Overall, the interplay between TPE and EV biology represents a promising frontier for both diagnostics and targeted plasma interventions.

## 8. Current Limitations and Future Iterations of TPE

### 8.1. Current Limitations

Among TPE limitations, vascular access site complications associated with TPE have been described previously (Table 1, Figure 1). For TPE there is also a need for specialized equipment and training, and in many hospital settings there is variable access to such equipment and properly trained staff. For example, though many ICUs have the necessary equipment, critical illness often overlays typical TPE indications which respond well to therapy [164,165,166]. Apheresis specialists, specialized coagulation guidelines, and nephrology teams may thus be required in many cases. With regard to cost, a recent Chilean cost comparison study indicated initial costs for a centrifugal TPE system range from USD $50,000–$60,000/device and USD $20,000–$25,000 for membrane filter systems [167], with prices/unit increasing towards $100,000 when including automated functions, and real-time monitoring capacity [168]. Annual maintenance is generally higher for membrane systems, running USD$1500–$2500/year with centrifugal TPE maintenance estimated at USD $600–$1000/year [167], which typically extends to slightly lower treatment costs/session for patients with centrifugal TPE [169]. Peripheral vascular access utilizes cheaper catheters, of which membrane filtration typically requires higher flow levels from central access points for bypassing filters, also contributing to cost differences [3,167].

In terms of patient treatment costs, these vary considerably, depending on setting (e.g., hospital vs. wellness clinic) and the complexity of the disease (e.g., specializing physician, ICU requirements, co-morbidities, etc.). Without insurance coverage, typical prices often fall somewhere between USD $1500 to >10,000/session [170,171,172,173]. Generally, TPE is conducted with a single nurse or apheresis specialist per session, under physician supervision [174]. Thus, salary considerations are often built into the overall cost models for TPE treatments. Though physicians are expected to have specialized expertise, the number of patients a single physician can supervise simultaneously is often influenced by institutional policies, staff levels, and the complexity of the disease or condition being treated.

Even at current costs, TPE is often a cost-effective alternative. For example, GBS and MG treatment by TPE is lower than intravenous immunoglobulin treatments [172]. Wider demand and routine usage on relatively healthy, normal aging populations should drive the costs of TPE sessions lower, especially when considering that age is the greatest predictor of most disease onset and repetitive TPE treatments appear to reduce biological age [175,176]. However, lowering costs to patients will likely require additional optimization and innovation. For example, alternative replacement fluids may exist such as hydroxyethyl starch instead of HSA, though there are concerns for critically ill patients [177,178], and innovations in synthetic blood and placement fluids are currently underway [179]. Revaluation of central access points and the implementation of small-volume exchanges for some conditions could also aid in lowering costs and patient accessibility [180]. Automating TPE through device innovations and real-time AI monitoring to aid supervising physicians may also lower future TPE costs/session. Lastly, clinical results which support the expansion of the number of conditions routinely treatable, especially age associated pathologies, may increase insurance coverage and reduce costs directly to patients. Demand continues to rise, from an estimated 43,000 apheresis collections within the US in 2021 to 132,000 in 2025 [168].

### 8.2. Future Iterations

Though we have extensively covered the TPE-mediated depletion of circulating drivers of disease, infusions of replacement plasma or therapies represent potential TPE adjuvants. That is, the condition of donor plasma may play an important role in defining therapeutic outcomes. Emerging evidence demonstrates that younger plasma improves the physical appearance and extends the lifespan of aged rodents [140] and further exerts an anti-inflammatory response in aged surgical patients [141]. However, the FDA has indicated there is no established clinical benefit for plasma infusions for aging, memory loss, or the treatment of heart disease, AD, or other neurological conditions [181]. Interestingly, exercise increases circulating EVs with miR-124 content and reduces apoptosis in cerebral ischemic-reperfusion injury mice [142,143]. Furthermore, albumin supplementation during TPE is necessary for maintaining blood vessel homeostasis, and preventing leaking into surrounding tissues, but functions also as a carrier of many pathological proteins (e.g., 95% of Aβ in AD patients) [43]. The removal of saturated albumin during TPE and supplementation with unbound albumin may thus also aid in sequestering toxic proteins and lipids. Refined TPE filters targeting removal of additional carriers of pathological content coupled with unsaturated and plasma conditioned replacement infusions may prove to be a powerful therapeutic paradigm.

The rapid evolution of high-throughput multi-omics has transformed our ability to interrogate circulating molecular signatures, and thus multi-omic diagnostics can guide TPE. Proteomics, epigenomics, transcriptomics, lipidomics, and metabolomics can be used to provide a comprehensive portrait of systemic health. Indeed, advancements in machine learning algorithms and artificial intelligence (AI) assisted network building have already yielded new biomarkers and actionable therapeutic disease targets in several disease contexts [144,145,146,147]. For example, a recent clinical trial on 42 individuals over the age of 50, receiving six total TPE sessions over three months with or without intravenous immunoglobulin therapy, demonstrated significant improvement in numerous biological age markers, in particular, across 15 different epigenetic “clocks” measuring biological age [175]. Three sessions reduced biological aging by 1.32 years, with immunoglobulin supplementation effectively doubling this age reduction [175]. Future data integration with other functional metrics of aging, such as motor skill improvements or memory recall, are essential for understanding which biomarkers of aging or disease effectively correlate with TPE-mediated therapeutic responses. Circulating tumor DNA (ctDNA), EV cargo, and single-cell transcriptomes from peripheral immune cells can reveal organ-specific dysfunction long before clinical symptoms arise [116,148]. The use of these diagnostic advancements coupled with real time monitoring of core signaling effects mediated by TPE should facilitate precision medicine and personalized patient specific plasma interventions, tuned for therapeutic responses. This would be further aided by longitudinal multi-omic profiling to identify patient responder subtypes and aid in the identification of mechanistic pathways through which TPE exerts its effects. Therefore, integrative computational platforms which synthesize “omic” data with clinical phenotypes may enable predictive and adaptive TPE, where exchange parameters, intervals, and durations are personalized based on molecular feedback loops.

## 9. Summary

Overall, current evidence for many TPE indications has been derived from a range of observational case studies, as is typical in reactionary medicinal fashion for established diseases. Hence, there is a pressing need for more prospective and randomized clinical trials to rigorously evaluate the efficacy and utility of various TPE applications in different diseases. To guide these trials, continued advancements in pre-clinical TPE modeling are needed (Figure 4) [44,45,113,114,182]. Mechanistically, TPE offers a unique modality for systemic ‘milieu reset’—a transient window in which inflammatory, metabolic, and vesicular noise is reduced. This depletion may temporarily lower systemic stress signals, tempering the immune system, and promoting stem cells, glia, and endothelium to re-engage reparative programs. While most ICUs or surgical units in developed healthcare systems/hospitals have well-established infrastructure, the availability of complete TPE facilities –including appropriate instruments, trained teams, and optimal replacement fluids—remain inadequate in many instances [166]. However, increasing the awareness of the broad range of therapeutic potential of TPE across diverse clinical indications, coupled with the training of highly skilled personnel dedicated to TPE administration, management, and monitoring could significantly enhance the clinical applicability and impact of TPE therapy [165,166].

## 10. Conclusions

When integrated with multi-omic diagnostics and deployed within a predictive framework, TPE has the potential to transition from an emergency intervention to a preventative and perhaps tissue rejuvenating modality. The temporal depletion of pathologic factors at least transiently alters metabolic, cardiovascular, and neuroinflammatory pathways, fundamentally altering the cellular, vesicular, and molecular composition of plasma. The simultaneous use of drugs or regenerative medicine products would be expected to operate differently during this window of plasma turnover. Given the strong history of protecting transplanted organs, it is reasonable to expect that other transplants (e.g., donor islets, HSCs, MSCs) may similarly benefit from TPE windows. Thus, future TPE initiatives should include combinatorial strategies pairing TPE with targeted biologics, regenerative EVs, and metabolic conditioning protocols. By integrating next-generation plasma diagnostics and precision medicine strategies, TPE has the potential to evolve into a foundational therapy for mitigating cellular stress and promoting long-term health across diverse diseases.

## Figures and Tables

**Figure 1 biomolecules-15-01000-f001:**
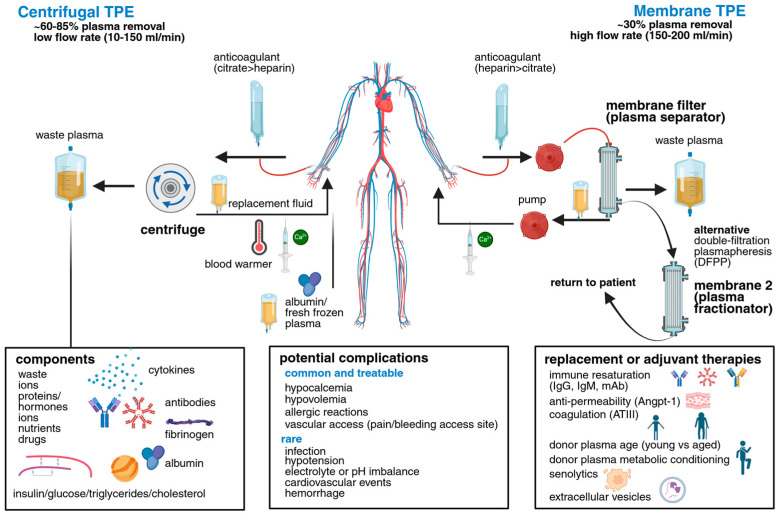
An overview of TPE strategies for clinical implications. Centrifugation TPE involves blood removal, anticoagulant addition typically favoring sodium citrate, and centrifuge-based plasma separation. Replacement albumin and fresh frozen donor plasma is infused upon fluid return to the patient, along with calcium supplementation prior to plasma return to avoid hypocalcemia. In membrane TPE patient blood (anticoagulated with heparin) is filtered over a hollow fiber filter, and fluids are returned as described for centrifugal TPE. Alternatively, in double-filtration plasmapheresis (DFPP), plasma separation is followed by a second membrane filter (plasma fractionator) to remove specific circulating components (e.g., autoantibodies) based predominately on size. Component depleted plasma is returned to patients. In general, membrane TPE utilizes a higher flow rate and is thus completed more quickly than centrifugal TPE. On the other hand, overall plasma removal is greater in centrifugal TPE per unit of time, with ~65–85% removal depending on donor exchange volume used for centrifugal, and ~30% for membrane filtration [2,3]. The use of either approach is typically indication specific. Common and rare complications are provided, of which most are due to either access (IV site pain) or rapid changes in blood volume, and easily treatable. Plasma components are highlighted, including those often implicated in human disease. Common and future replacement or adjuvant therapies are also emphasized and discussed further in subsequent sections. Created in BioRender. Tompkins, J. (2025) https://BioRender.com/5r1lysi, accessed on 6 June 2025.

**Figure 2 biomolecules-15-01000-f002:**
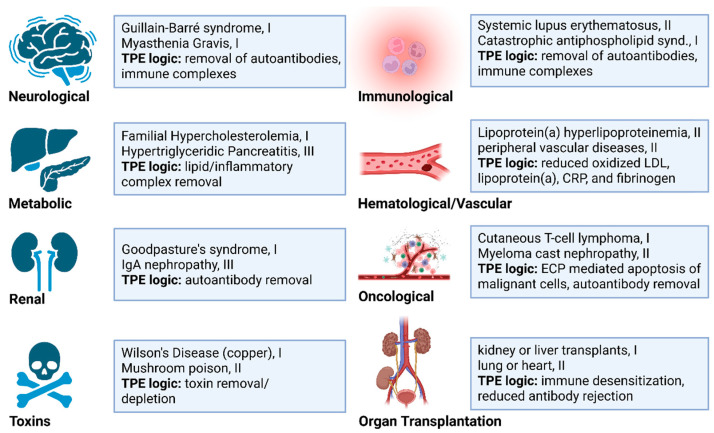
Current disease-specific indications and applications of TPE in clinical practice. Two example indications are shown for each disease category, followed by category ranking from ASFA guidelines [9]. I = Disorders for which apheresis is accepted as first-line therapy. II = Disorders for which apheresis is accepted as second-line therapy. III = Optimum role of apheresis therapy is not established. ECP = extracorporeal photopheresis. Created in BioRender. Tompkins, J. (2025) https://BioRender.com/7d5zyiz, accessed on 6 June 2025.

**Figure 3 biomolecules-15-01000-f003:**
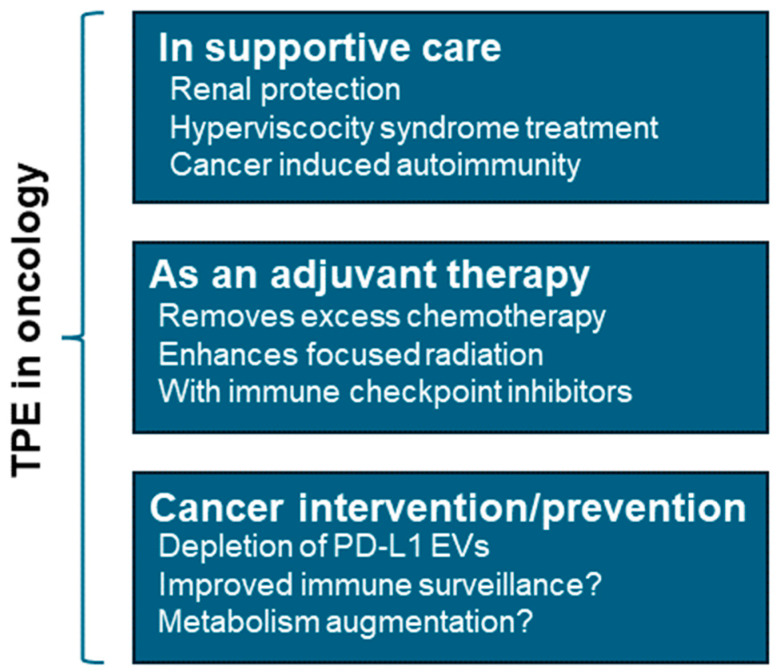
An overview of TPE applications in oncology. TPE roles in supportive involve removal of damaging associated immune complexes, reducing hyperviscosity syndrome, cancer-associated autoimmunity, and aiding in renal protection. As an adjuvant, evidence points to synergistic applications for TPE in combating adverse immune related events to checkpoint inhibitor treatments with radiation [87]. TPE effectively removes many excess chemotherapies, which may enable increasing doses in future trials. As a cancer intervention, the depletion of tumor driving PD-L1 may reduce tumor recurrence. Underexplored areas of research involve systemic immune system and metabolic augmentation to those less favorable for cancer development.

**Figure 5 biomolecules-15-01000-f005:**
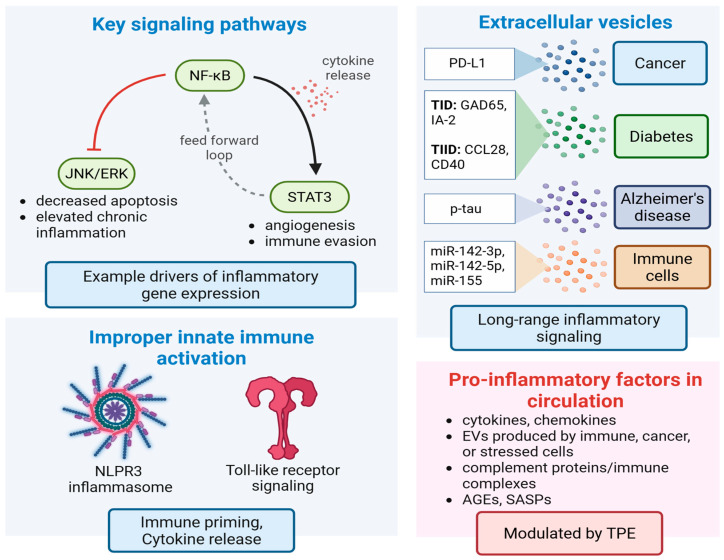
Signaling pathways, EVs, and inflammatory signaling potentially altered by TPE. An overview of key signaling categories contributing to chronic systemic inflammation: (1) transcription factor pathways (NF-κB, STAT3, MAPK/JNK/ERK) that regulate inflammatory gene expression and immune evasion; (2) innate immune sensors (TLRs, NLRP3 inflammasome) driving release of cytokines and chemokines (e.g., IL-1, IL-6, TNF-α); (3) extracellular vesicles (EVs) produced by immune, cancer, or stressed cells that carry pro-inflammatory cargo (e.g., PD-L1, p-tau, autoantigens, miRNAs). Therapeutic plasma exchange (TPE) reduces inflammatory burden by depleting cytokines, immune complexes, and EVs, promoting immune modulation (e.g., increased Tregs, Th2 shift) and resulting in a more regenerative systemic environment. Created in BioRender. Hussey, D, Tompkins J. https://BioRender.com/tqvmd79, accessed on 6 June 2025.

**Table 2 biomolecules-15-01000-t002:** Neuroimmune interfaces with differential exposure to circulating plasma factors potentially altered by TPE.

Neuroimmune Interface	Access Potential *	Key Cell Types	Plasma Factors Implicated	Ref.
BBB Intact	Low	Endothelial cells, pericytes, astrocyte endfeet	CCL11, IL-6, TNFα	[46,47,48,49]
BBB (aged, dysfunctional)	Moderate	Endothelial cells, microglia, neurons	β2-micoglobulin, oxidized LDL, fibrinogen	[32,44,45]
Choroid Plexus (Blood-CSF barrier)	High	Epithelial cells, infiltrating T cells, macrophages	IFN-γ, IL1β, CXCL10	[41,50]
Meninges	High	Dura-resident macrophages, B/T cells, dendritic cells	CXCL12, VEGF, IL-6	[48,51]

* “Access potential” refers to the extent to which circulating molecules affected by TPE can influence each interface, either by direct diffusion or indirect immune signaling. Supporting references correspond to the cited role of specific cytokines, hormones, or structural properties of each barrier.

## Data Availability

The raw HLI data supporting the conclusions of this article will be made available by the authors on request to corresponding author.

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
