# Peer review of "Therapeutic Plasma Exchange: Current and Emerging Applications to Mitigate Cellular Signaling in Disease"

_biomolecules, 2025, doi:10.3390/biom15071000_

Round 1
Reviewer 1 Report
Comments and Suggestions for Authors
The manuscript provides a timely review of therapeutic plasma exchange, including its mechanisms, current clinical applications, and emerging research directions across cardiovascular, metabolic, neurological, and oncological diseases. The review is well-structured, with clear sections. However, several areas could be enhanced to strengthen the manuscript.
1.While TPE’s role in removing PD-L1+ EVs and chemotherapeutic agents is noted, the discussion of cancer metabolism (e.g., Warburg effect) is superficial. Elaborate on how TPE modulates metabolic pathways in tumor cells or the tumor microenvironment.
2.The automated syringe pump-based exchange platform (Figure 3) is innovative, but clarify how this model mimics human TPE (e.g., plasma volume replaced, frequency of exchanges). Discuss whether murine findings (e.g., improved hind limb ischemia recovery) translate to humans, given species-specific differences in immune and vascular biology.
3.TPE’s limitations (e.g., access site complications, need for specialized equipment, cost) are mentioned briefly in Figure 1 but deserve a dedicated section. Discuss how these barriers impact widespread adoption, especially in resource-limited settings.
4.The manuscript emphasizes personalized TPE but lacks specifics on validated biomarkers (e.g., which EV cargoes or cytokines predict response). Cite recent omics studies that could inform biomarker discovery.
Author Response
Comment 1: The manuscript provides a timely review of therapeutic plasma exchange, including its mechanisms, current clinical applications, and emerging research directions across cardiovascular, metabolic, neurological, and oncological diseases. The review is well-structured, with clear sections. However, several areas could be enhanced to strengthen the manuscript.
- While TPE’s role in removing PD-L1+ EVs and chemotherapeutic agents is noted, the discussion of cancer metabolism (e.g., Warburg effect) is superficial. Elaborate on how TPE modulates metabolic pathways in tumor cells or the tumor microenvironment.
Response 1: We agree with this comment and have elaborated on how TPE modulates metabolic pathways in tumor cells or the tumor microenvironment. Specifically, in the section Cancer metabolism and TPE we have expanded on the Warburg effect, noting the function of acidosis and lactate in supporting cancer growth and immune evasion with relevant recent citations. We further note “TPE was shown to significantly reduce lactic acid levels in septic shock patients at 24h and significantly improved patient (Keith et al, 2020). Thus, it is possible that routine TPE treatments could mitigate tumor acidosis.” We have restructured sentences a bit in this section and conclude that “metabolic mitigation of cancer development and metastasis through TPE represents an important treatment paradigm for researcher exploration, especially for TPE mediated effects within tumor microenvironment.” Lastly, we added a sentence in TPE as a cancer intervention on the role of PD-L1 in suppressing T cell function (first paragraph).
Comment 2: The automated syringe pump-based exchange platform (Figure 3) is innovative, but clarify how this model mimics human TPE (e.g., plasma volume replaced, frequency of exchanges). Discuss whether murine findings (e.g., improved hind limb ischemia recovery) translate to humans, given species-specific differences in immune and vascular biology.
Response 2: We thank the reviewer for this suggestion and have added a description of how this pre-clinical model compares with human therapies, including volumes of exchange and timing of procedures, and we have discussed species-specific differences in vascular biology to the section Preclinical modeling of TPE. We have also added in descriptions for and new references for a rat TPE device and acoustic fluidic device for small volume apheresis in this section.
Comment 3: TPE’s limitations (e.g., access site complications, need for specialized equipment, cost) are mentioned briefly in Figure 1 but deserve a dedicated section. Discuss how these barriers impact widespread adoption, especially in resource-limited settings.
Response 3: We agree with this suggestion and have addressed this in two ways. First, we added in a new Table 1 which includes known complications. Secondly, we have retitled section 7 (now section 8) to “Current Limitations and Future Iterations of TPE,” and now include a comprehensive section specifically addressing TPE limitations (8.1) including access site complications, need for specialized equipment, cost estimates, and opportunities to reduce costs.
Comment 4: The manuscript emphasizes personalized TPE but lacks specifics on validated biomarkers (e.g., which EV cargoes or cytokines predict response). Cite recent omics studies that could inform biomarker discovery.
Response 4: We agree that personalized TPE strategies would benefit from the knowledge of which disease associated biomarkers are sustainably depleted by TPE, of which this knowledge is often lacking in current literature, at least for new potential TPE indications. Circulating biomarkers are provided for numerous diseases throughout the manuscript and in a few places we now emphasize direct correlations between a biomarker and TPE benefit, or the lack of data to demonstrate durable depletion. One specific example includes paragraph two of section 7 “EVs in metabolic disease.” The final sentence now concludes with …”however, the durability of these effects remains to be determined.” Another example includes section 7 “EVs in tumor progression”, second to last sentence. This sentence now reads “These observations strongly argue for TPE in contemporary cancer therapy regimes, aimed at reducing tumor driving EVs, especially those containing well established and TPE depletable disease biomarkers such as PD-L1 (Figure 5).
In the section 8.2 Future Iterations, we provide a new example of -omic data which has been mined for specific novel circulating biomarkers and emphasize that TPE durability and therapeutic responses also need to be taken into consideration. We specifically expand on a recent clinical trial involving several epigenetic “clocks” which measure improvements to biological aging with six TPE sessions (Fuentealba, et al, 2025). A couple of sentences later, we have note “The use of these diagnostic advancements coupled with real time monitoring of core signaling effects mediated by TPE should facilitate precision medicine…tuned for therapeutic responses.”
Reviewer 2 Report
Comments and Suggestions for Authors
The manuscript presents a thorough, well-structured, and thoughtfully compiled review. I would like to offer a few suggestions and technical comments.
1) General remark: It would be beneficial to clearly distinguish between preclinical (animal model), human observational, and human randomized data.
An Overview of Plasma and TPE Technologies
2) Certain diseases (e.g., CAPS, SLE) are only briefly mentioned. It would be helpful to expand on the pathophysiology or provide brief treatment algorithms.
4.1. Metabolic disease
3) There is limited human data on T2D. It would be worth noting that, at present, this indication remains theoretical. A short comment on what evidence would be required for it to reach "Category III" status would be useful.
6.1. Cancer
4) The section "removing excess chemotherapy" conflates the treatment of acute toxic exposures with long-term drug elimination. TPE is not suitable for removing all chemotherapeutic agents—its efficacy is limited to high-molecular-weight, plasma protein-bound compounds (e.g., monoclonal antibodies, cisplatin). This point could be supported with pharmacokinetic examples.
5) The section on "Cancer metabolism and TPE" does not provide substantial explanation of how TPE alters tumor metabolism. The mention of the Warburg effect feels somewhat formulaic. At least one in vitro or in vivo study should be cited, showing how post-TPE plasma changes impact tumor cell metabolism or proliferation. The Warburg effect remains an intensely studied phenomenon in tumor biology and cell lines—this literature, including potential therapeutic implications, should be incorporated.
Editing and Formatting Future manuscripts should be edited more carefully. The topic of the review is relevant and well thought-out, but structural coherence has been neglected. Readers must be able to absorb the presented information effectively. Unfortunately, the unedited and unchecked format currently overshadows the scientific content.
6) Figures are crowded, and the main text lacks adequate explanations or references to them.
7) References are frequently incomplete or missing.
8 A) Figure 1: The title is duplicated ("An overview of TPE strategies").
8 B) Figure 2: Again, the title is repeated.
8 C) Figure 3: Same issue. Why are the titles identical?
9) The section numbering needs to be revised. For example, there are two sections labeled 6.1 (among other issues). Chapter 3 also contains significant structural errors. I attempted to highlight these, but even my Word processor got confused 😊. This needs to be corrected.
10) 1.1. "Current metabolic disease indications for TPE" – This is a misleading title within a section on neurodegenerative diseases. It is likely a typo and should be labeled “neurologic disease” or “neuroimmune disease.”
Author Response
Comment 1: The manuscript presents a thorough, well-structured, and thoughtfully compiled review. I would like to offer a few suggestions and technical comments.
1) General remark: It would be beneficial to clearly distinguish between preclinical (animal model), human observational, and human randomized data.
Response 1: We agree and have made several updates within the manuscript to clearly define whether a human study is observational or randomized. Further in section 3, “New Disease Targets with Potential Therapeutic Reponses to TPE,” we end the first paragraph with the following sentence: “All studies described are conducted in humans, unless otherwise indicated for preclinical work.”
Comment 2: Certain diseases (e.g., CAPS, SLE) are only briefly mentioned. It would be helpful to expand on the pathophysiology or provide brief treatment algorithms.
Response 2: We agree that these designations are helpful. To address, we have provided a new Table 1 describing typical TPE numbers and frequency, concurrent therapies, benefits, and associated potential complications. Additionally, the first sentence of paragraph four in section 2 “Diseases Currently Benefiting from Therapeutic Apheresis” now reads: “For a list and description of all current therapeutic apheresis indications, ranked by category (Category I–III) of disease responsiveness, readers are referred to Connelly-Smith et al., 2023.”
Comment 3:
4.1. Metabolic disease
There is limited human data on T2D. It would be worth noting that, at present, this indication remains theoretical. A short comment on what evidence would be required for it to reach "Category III" status would be useful.
Response 3: In the text, we more clearly acknowledge current limitations to treating DM. For example, paragraph two of section 7 “EVs in metabolic disease.” The final sentence now concludes with …”however, the durability of these effects (TPE) remains to be determined.” In “Emerging TPE applications in metabolic disease” section, we have added the following text: “With recent developments in preclinical modeling of TPE (Figure 3, and larger randomized clinical trials focused on metabolic inclusion criteria, TPE may be-come important for improving some T2D outcomes.” We further provide a new section, “8.1 Current Limitations”, which describes TPE costs and potential cost optimization approaches. Lastly, evidence suggests some instances a few or several TPE sessions can have significant effects in aging. In section “8.2 Future Iterations” regarding omic-scale studies which have demonstrated significant reductions to biological age, as measured by several epigenetic clocks, with as few as three TPE sessions (paragraph 2, sentences 3-5), and provided relevant citations. Reductions in biological age are expected to aid in delaying the onset of both AD and metabolic disease.
Comment 4:
6.1. Cancer
The section "removing excess chemotherapy" conflates the treatment of acute toxic exposures with long-term drug elimination. TPE is not suitable for removing all chemotherapeutic agents—its efficacy is limited to high-molecular-weight, plasma protein-bound compounds (e.g., monoclonal antibodies, cisplatin). This point could be supported with pharmacokinetic examples.
Response 4: In this section we more clearly refer to acute cisplatin toxicity and refer readers to a review paper by Mahmoud et al, which describes pharmacokinetic differences.
Comment 5: The section on "Cancer metabolism and TPE" does not provide substantial explanation of how TPE alters tumor metabolism. The mention of the Warburg effect feels somewhat formulaic. At least one in vitro or in vivo study should be cited, showing how post-TPE plasma changes impact tumor cell metabolism or proliferation. The Warburg effect remains an intensely studied phenomenon in tumor biology and cell lines—this literature, including potential therapeutic implications, should be incorporated.
Response 5: We agree with the comment, also raised by reviewer 1. This response is provided here as well: We agree with this comment and have elaborated on how TPE modulates metabolic pathways in tumor cells or the tumor microenvironment. Specifically, in the section Cancer metabolism and TPE we have expanded on the Warburg effect, noting the function of acidosis and lactate in supporting cancer growth and immune evasion with relevant recent citations. We further note “TPE was shown to significantly reduce lactic acid levels in septic shock patients at 24h and significantly improved patient survival. Thus, it is possible that routine TPE treatments could mitigate tumor acidosis.” We have restructured sentences a bit in this section and conclude that “metabolic mitigation of cancer development and metastasis through TPE represents an important treatment paradigm for researcher exploration, especially for TPE mediated effects within tumor microenvironment.” Lastly, we added a sentence in TPE as a cancer intervention on the role of PD-L1 in suppressing T cell function (first paragraph).
Comment 6: Editing and Formatting Future manuscripts should be edited more carefully. The topic of the review is relevant and well thought-out, but structural coherence has been neglected. Readers must be able to absorb the presented information effectively. Unfortunately, the unedited and unchecked format currently overshadows the scientific content.
Response 6: We understand these concerns and have made several structural changes to improve clarity, paragraph structure, and clarity. Additionally, we have had an internal grants expert review the manuscript for these purposes.
Comment 7: Figures are crowded, and the main text lacks adequate explanations or references to them.
Response 7: Additional references to figures in the main text of the article have been included. Figures have been edited for spacing and clarity where possible.
Comment 8: References are frequently incomplete or missing.
Response 8: Additional references have been added throughout the main text.
Comment 9:
A) Figure 1: The title is duplicated ("An overview of TPE strategies").
B) Figure 2: Again, the title is repeated.
C) Figure 3: Same issue. Why are the titles identical?
Response 9: These typographical errors have been fixed and each figure title adjusted.
Comment 10: The section numbering needs to be revised. For example, there are two sections labeled 6.1 (among other issues). Chapter 3 also contains significant structural errors. I attempted to highlight these, but even my Word processor got confused ?. This needs to be corrected.
Response 10: Thank you for catching these mistakes. We have corrected section numbering throughout and improved the structure of chapter 3 in particular.
Comment 11: 1.1. "Current metabolic disease indications for TPE" – This is a misleading title within a section on neurodegenerative diseases. It is likely a typo and should be labeled “neurologic disease” or “neuroimmune disease.”
Response 11: This was indeed a typographical error. The title has been changed to “Current neurodegenerative disease indications for TPE.”
Reviewer 3 Report
Comments and Suggestions for Authors
Rony et al. present a review of the role played by therapeutic plasma exchange (TPE) on cellular signaling in aging and disease. The rationale for the manuscript is to raise awareness of the established and forward thinking roles TPE plays in the treatment of disease.
The authors present a brief history of the genesis of TPE first in animal models as a form of dialysis/kidney replacement, followed by its well know role as a method to remove antibodies that mediate GBS and MG. Subsequently, the role of TPE in modifying neurological, metabolic, cancer related, and other diseases are presented in generally well-written sections wherein signaling molecules or EVs containing key pathological molecules can be removed and improve or at least modify the trajectory of several illnesses.
While TPE may improve some aspects of metabolic disease, this reviewer finds it difficult to imagine TPE playing much of a role in the treatment of DM given the sheer expense of TPE and the huge number of the population afflicted with DM. It seems to me that TPE modulation of states that involve short-term release of pathological antibodies or other macromolecules that act as cellular signals makes more sense from an economical standpoint. The same is true of disease such as AD. If a few TPE sessions could significantly delay or abrogate the debilitating neurological effects of for a significant period of time, great. Otherwise, there is not a sustainable medical business model possible.
Amongst the myriad of diseases mentioned, there was one very important disease not mentioned in the cardiovascular section that I think is worth including. Heparin induced thrombotic thrombocytopenia (HITT) is a huge problem that is growing with our aging population and concordant increase in exposure to heparin. I have seen TPE administered before and after heart transplantation and other procedures requiring cardiopulmonary bypass. I have included a literature search as a PDF attachment for the authors’ convenience. While the use of heparin alternatives have been advocated, I have seen my share of patients have clotting on pump or severe coagulopathy after bypass. Many practitioners prefer to remove antibodies, administer heparin, and repeat plasmapheresis to maximize patient safety.
I have only two more comments.
Paragraph layout. Sometimes paragraphs are oddly split between figures, and sometimes a new paragraph indent occurs in the middle of a sentence from the previous paragraph. The authors need to carefully correct their layout.
Illustrations. It would be appreciated if the authors increased either the size of the font or overall size of the figures to make them easier to read and interpret.

Author Response
Comment 1: Rony et al. present a review of the role played by therapeutic plasma exchange (TPE) on cellular signaling in aging and disease. The rationale for the manuscript is to raise awareness of the established and forward thinking roles TPE plays in the treatment of disease.
The authors present a brief history of the genesis of TPE first in animal models as a form of dialysis/kidney replacement, followed by its well know role as a method to remove antibodies that mediate GBS and MG. Subsequently, the role of TPE in modifying neurological, metabolic, cancer related, and other diseases are presented in generally well-written sections wherein signaling molecules or EVs containing key pathological molecules can be removed and improve or at least modify the trajectory of several illnesses.
While TPE may improve some aspects of metabolic disease, this reviewer finds it difficult to imagine TPE playing much of a role in the treatment of DM given the sheer expense of TPE and the huge number of the population afflicted with DM. It seems to me that TPE modulation of states that involve short-term release of pathological antibodies or other macromolecules that act as cellular signals makes more sense from an economical standpoint. The same is true of disease such as AD. If a few TPE sessions could significantly delay or abrogate the debilitating neurological effects of for a significant period of time, great. Otherwise, there is not a sustainable medical business model possible.
Response 1: We appreciate the reviewer's point of views concerning the clinical applicability of TPE in a real-world scenario, particularly for diseases such as DM or AD which implicate many patients with varying degree of disease severity and/or financial conditions. In the text, we more clearly acknowledge current limitations to treating DM. Specifically…
We further provide a new section, “8.1 Current Limitations”, which describes TPE costs and potential cost optimization approaches. Lastly, we do contend that in some instances a few or several TPE sessions can have significant effects in aging and AD circumstances. In particular we cite the AMBAR trial, which shows significant delays to cognitive and functional decline in AD, and have updated the text in section 3, “Emerging neuroimmune TPE indications”, paragraph two to include trial details of six TPE sessions and monitoring for a year. We further have added details in section “8.2 Future Iterations” regarding omic-scale studies which have demonstrated significant reductions to biological age, as measured by several epigenetic clocks, with as few as three TPE sessions (paragraph 2, sentences 3-5), and provided relevant citations. Reductions in biological age are expected to aid in delaying the onset of both AD and metabolic disease.
Comment 2: Amongst the myriad of diseases mentioned, there was one very important disease not mentioned in the cardiovascular section that I think is worth including. Heparin induced thrombotic thrombocytopenia (HITT) is a huge problem that is growing with our aging population and concordant increase in exposure to heparin. I have seen TPE administered before and after heart transplantation and other procedures requiring cardiopulmonary bypass. I have included a literature search as a PDF attachment for the authors’ convenience. While the use of heparin alternatives have been advocated, I have seen my share of patients have clotting on pump or severe coagulopathy after bypass. Many practitioners prefer to remove antibodies, administer heparin, and repeat plasmapheresis to maximize patient safety.
Response 2: We have incorporated a new section discussing HIT pathology, and recent studies highlighting the potential of TPE in reducing HIT autoantibodies during cardiac surgery and limiting HIT-associated mortality. We believe this section strengthens the clinical potential of TPE.
Comment 3: Paragraph layout. Sometimes paragraphs are oddly split between figures, and sometimes a new paragraph indent occurs in the middle of a sentence from the previous paragraph. The authors need to carefully correct their layout.
Response 3: We agree with this critique and have carefully formatted and revised sentence and paragraph structures throughout to improve clarity and readability.
Comment 4: Illustrations. It would be appreciated if the authors increased either the size of the font or overall size of the figures to make them easier to read and interpret.
Response 4: We agree with this suggestion and have adjusted spacing, fonts, and figure sizes to improve readability.
Reviewer 4 Report
Comments and Suggestions for Authors
This is a relatively thorough compilation of TPE’s possible clinical benefits. The main shortcoming of the manuscript is that it lacks discussions of the authors own clinical suggestions that could make the review having a point of nuance.
TPE is generally a temporary treatment strategy, and needs to be repeated in a certain interval based on individual clinical treatment. Therefore, questions such as: for each cited study, what is the longitudinal time line? What are the side effects due to prolonged TPE? Under a given condition, does TPE function as the main treatment strategy or in parallel with another medical intervention? In addition, what are the current clinical practices that uses TPE as their primary treatment, is it Kidney diseases? CVD patients? How accessible is TPE in ICU setting when patients might benefit immediately? What is your suggestion?
Author Response
Comment 1:
TPE is generally a temporary treatment strategy, and needs to be repeated in a certain interval based on individual clinical treatment. Therefore, questions such as: for each cited study, what is the longitudinal time line? What are the side effects due to prolonged TPE? Under a given condition, does TPE function as the main treatment strategy or in parallel with another medical intervention? In addition, what are the current clinical practices that uses TPE as their primary treatment, is it Kidney diseases? CVD patients? How accessible is TPE in ICU setting when patients might benefit immediately? What is your suggestion?
Response 1:
We appreciate these important questions and have addressed these questions within the revised manuscript. In particular, we have provided a new Table 1 with descriptions of the number and frequency of TPE sessions for known indications, as well as concurrent therapies, and potential complications. Additional TPE treatment designs are now included for Alzheimer’s studies and for age intervention under section 8.2 Future Iterations. Lastly, we have included a new section 8.1, which specifically details TPE limitations and associated costs. In this section we touch on the need for specialized equipment and how critical illness in the ICU often overlay primary TPE indications. In the final paragraph of this section we provide suggestions to reduce costs and improve routine TPE use. Lastly, In the summary section (first paragraph, final 2 sentences) of our revised manuscript, we addressed the current limitations of TPE implementation in ICU settings and propose that healthcare institutions should be equipped with comprehensive TPE facilities and appropriately trained to ensure timely and effective access of TPE therapy to critically ill patients.
Reviewer 5 Report
Comments and Suggestions for Authors
The study is timely, very interesting and well-organized. However, I think that before a final decision is made, some issues should be addressed. Therefore, i propose you to make some changes in order to render it more attractive.
Major comments
Lines 38-40: In the introduction you mentioned only GBS but I think PLEX should be mentioned above all for all neurological steroid-resistants conditions requiring additional therapies. In this context, I’m referring to neurological immune related toxicities (n-iraes; please see Malvaso et al., Brain Sciences (2024) for a review and add some relevant informations/referens in order to cover more details about PLEX and neurological conditions). Other relevant references are missing and introduction is presented in an historical form. You should reformat it to be more scientifically sound. I report you some other reference in my opinion essential to add contents and details:
-Rossi et al., 2024 Neurology, Neuroimmunology and Neuroinflammation
You can add and integrate, if possible, these informations also in lines 104-108, Table 1 and 3.1.1 paragraph and 6.1.2 paragraph.
Lines 605-607: you reported that plasma depletion may be associated with observations of increased murine hippocampal proliferation, improving learning and memory. Therefore, as for neurological conditions extremely related to hippocampal damage (e.g., autoimmune encephalitis), do you have data about the rationale of the use of PLEX in this condition. For example, a recent systematic review reported for the first time results about a new isolated cognitive deficit in patients with LGI1/CASPR2 autoimmune encephalitis. In my opinion could be interesting to study and report if possible an eventual rationale with usefulness of PLEX in the memory disturbances in the clinical context of autoimmune encephalitis (please see Malvaso et al., 2025 European Journal of Neurology).
Generally, my only real criticism is to the structure of the paragraph, they are too many, i suggest you to revise the text trying to clean it from not relevant informations.
Minor comments:
- Please diuble check for typos
- Please revise English Language in the whole manuscript
- Be careful in the division of the paragraph, avoiding boring reading.
- Figures are well depicted
- Tables should be implemented with informations I suggested above.
Minor revision of the English language
Author Response
Comment 1:
The study is timely, very interesting and well-organized. However, I think that before a final decision is made, some issues should be addressed. Therefore, i propose you to make some changes in order to render it more attractive.
Major comments
Lines 38-40: In the introduction you mentioned only GBS but I think PLEX should be mentioned above all for all neurological steroid-resistants conditions requiring additional therapies. In this context, I’m referring to neurological immune related toxicities (n-iraes; please see Malvaso et al., Brain Sciences (2024) for a review and add some relevant informations/referens in order to cover more details about PLEX and neurological conditions). Other relevant references are missing and introduction is presented in an historical form. You should reformat it to be more scientifically sound. I report you some other reference in my opinion essential to add contents and details:
-Rossi et al., 2024 Neurology, Neuroimmunology and Neuroinflammation
You can add and integrate, if possible, these informations also in lines 104-108, Table 1 and 3.1.1 paragraph and 6.1.2 paragraph.
Response 1:
We have removed the example of GBS in this introductory paragraph and integrated this information and suggested references into section 2 (Diseases Currently Benefiting from Therapeutic Apheresis) and in section 3.1.1. Section 6.1 (now section 7.1) has to do with Alzheimer’s disease and is not relevant to the suggested reference.
We feel that a historical introduction is informative, appreciated by the wide readership of Biomolecules, and have kept this material in the manuscript.
Comment 2:
Lines 605-607: you reported that plasma depletion may be associated with observations of increased murine hippocampal proliferation, improving learning and memory. Therefore, as for neurological conditions extremely related to hippocampal damage (e.g., autoimmune encephalitis), do you have data about the rationale of the use of PLEX in this condition. For example, a recent systematic review reported for the first time results about a new isolated cognitive deficit in patients with LGI1/CASPR2 autoimmune encephalitis. In my opinion could be interesting to study and report if possible an eventual rationale with usefulness of PLEX in the memory disturbances in the clinical context of autoimmune encephalitis (please see Malvaso et al., 2025 European Journal of Neurology).
Response 2:
We agree with this suggestion have added the suggested citation and referred to potential treatment of autoimmune encephalitis and cognitive deficit treatment in section 6, paragraph 1.
Comment 3:
Generally, my only real criticism is to the structure of the paragraph, they are too many, i suggest you to revise the text trying to clean it from not relevant informations.
Response 3:
The manuscript has been edited throughout for sentence clarity and paragraph structure, with several merged as appropriate, and vetted by co-authors.
Comment 4:
Minor comments:
- Please diuble check for typos
- Please revise English Language in the whole manuscript
- Be careful in the division of the paragraph, avoiding boring reading.
- Figures are well depicted
- Tables should be implemented with informations I suggested above.
Response 4: Again, the manuscript has been edited throughout for sentence clarity and paragraph structure. Additionally, we have had an internal grants expert review the manuscript for proper English usage and format. We are grateful for this review of our manuscript, but with respect, the English used by this reviewers’ critiques is itself improper and filled with typological errors. Further the remaining four reviewers indicated that the English is fine
Round 2
Reviewer 3 Report
Comments and Suggestions for Authors
The authors have addressed my concerns. I have no further comments.